# Gender Difference in Fear and Anxiety about and Perceived Susceptibility to COVID-19 in the Third Wave of Pandemic among the Japanese General Population: A Nationwide Web-Based Cross-Sectional Survey

**DOI:** 10.3390/ijerph192316239

**Published:** 2022-12-04

**Authors:** Rio Sasaki, Atsuhiko Ota, Hiroshi Yatsuya, Takahiro Tabuchi

**Affiliations:** 1Department of Public Health, Fujita Health University School of Medicine, Toyoake 470-1192, Japan; 2Department of Public Health and Health Systems, Nagoya University Graduate School of Medicine, Nagoya 466-8550, Japan; 3Cancer Control Center, Osaka International Cancer Institute, Osaka 541-8567, Japan

**Keywords:** anxiety, COVID-19, fear, gender difference, infectious disease pandemic, perceived susceptibility

## Abstract

Existing research suggested gender differences in fear and anxiety about and perceived susceptibility to COVID-19 and previous infectious disease pandemics. We analyzed whether women felt fear and anxiety about and perceived susceptibility to COVID-19 more frequently than men in Japan. We conducted a cross-sectional analysis using internet survey data collected during the third wave of the pandemic in Japan. The subjects were enrolled from the Japanese general population: 11,957 men and 11,559 women. Fear and anxiety specifically related to COVID-19 were evaluated with the Japanese version of the Fear of COVID-19 Scale (FoCS). The question “How likely do you think you will be infected with COVID-19?” was used to assess the perceived susceptibility to COVID-19. Women had higher mean (standard deviation) FoCS scores [18.6 (5.6) vs. 17.5 (5.9), d = 0.190] and reported the median or higher FoCS score (57.4% vs. 51.4%, φ = 0.060) and perceived susceptibility (13.6% vs. 11.5%, φ = 0.032) more frequently than men. The odds ratios (95% confidence intervals) adjusted for age, having a spouse, comorbidities, watching commercial TV stations’ news programs, employment status, and household income were 1.24 (1.17–1.32) and 1.27 (1.16–1.38), respectively. We observed that women were more anxious and fearful about and perceived the susceptibility to infectious diseases more frequently than men even one year after the pandemic occurred in Japan, although the effect size was small.

## 1. Introduction

Women were reported to feel fear and anxiety about COVID-19 more frequently than men in many countries, such as the U.S. [1,2], China [3], and Pakistan [4]. Researchers developed questionnaires to assess the fear and anxiety specifically related to COVID-19, such as the Fear of COVID-19 Scale (FoCS) [5] and the Anxiety and Fear of COVID-19 (AMICO) Assessment Scale [6]. In Brazil [7] and Spain [8], women had higher FoCS scores than men. In Japan, Midorikawa et al. reported a similar finding based on data collected in early August 2020 [9]. In Spain, women had higher AMICO scores than men [10]. A similar tendency was found in previous infectious disease pandemics. In Asian countries, women felt fear and anxiety more frequently than men about the pandemic of severe acute respiratory syndrome (SARS) [11,12] and Middle East respiratory syndrome (MERS) [13].

Unlike fear and anxiety, perceived susceptibility may not show the prominent gender difference. Women were reported to perceive greater susceptibility to COVID-19 more frequently than men in the U.S. [14] and Iran [15] but not in Israel [16,17]. The inconsistency was found in the previous infectious disease pandemics. In the 2003 SARS outbreak, women were more worried about contracting SARS for themselves than men in Hong Kong [11] and the Netherlands [18]. In the outbreak of avian influenza in Asia, women perceived more susceptibility than men in Korea [19] but not in Hong Kong [20]. No study addressed the gender difference in perceived susceptibility to COVID-19 in Japan, to our knowledge.

Gender difference does not merely mean biological differences. Socioeconomic statuses (marriage rate, employment rate/status, income, etc.) [21], health statuses such as the prevalence of non-communicable diseases (NCDs) [22], and lifestyles (exposure to media, etc.) [23] differ between men and women in Japan. These factors must be considered simultaneously when examining the effects of gender differences. In this study, considering these factors, we analyzed whether women felt fear and anxiety about and perceived susceptibility to COVID-19 more frequently than men in Japan.

## 2. Materials and Methods

### 2.1. Study Design

A cross-sectional study was conducted.

### 2.2. Subjects

We used data from a follow-up survey of the Japan Society and New Tobacco Internet Survey (JASTIS) that was conducted between 8 and 26 February 2021: we call it the JASTIS 2021 hereafter [24,25]. The first COVID-19 case was found in January 2020 and the third wave occurred between November 2020 and March 2021. The JASTIS is an internet-based large-scale cohort study. It was launched in 2015. The original aim was to describe the usage of new tobacco products and their related factors in Japan.

In response to the COVID-19 pandemic, the JASTIS collaborated with the Japan COVID-19 and Society and New Tobacco Internet Survey (JACSIS). The JACSIS was launched in 2020 and aimed to evaluate the health conditions and social determinants of the COVID-19 pandemic in Japan [26,27]. The baseline examination of the JACSIS (JACSIS 2020) was conducted between 25 August and 30 September 2020, i.e., amid the second wave of the pandemic in Japan. The JACSIS 2020 was a self-administered questionnaire survey using a survey panel with approximately 2.2 million individuals having diverse socioeconomic backgrounds, including educational level, household income, the number of household members, etc. We called for approximately 224 thousand eligible participants who were stratified by gender, age, and prefecture from the panel to ensure national representation. The enrolment continued until the target number of respondents was reached. We made the target number 28,000 (12.5% of the eligible participants) and a priori set regarding the age, gender, and prefectures, referring to the distribution of the general Japanese population in 2019.

For the JASTIS 2020, we called for the JACSIS 2020 participants and other survey panelists who were stratified by gender, age, and prefecture like we did for the JACSIS 2020. The enrolment continued until the target number of respondents, 26,000, was reached.

We set an item “Choose the penultimate option from the following five options” in the questionnaire to extract the participants who responded honestly to the questionnaire with a correct understanding of language. Those who did not correctly answer the item (*n* = 2484, 9.6%) were excluded. Finally, the remaining 23,516 respondents, i.e., 11,957 men and 11,559 women, were analyzed in this study.

### 2.3. Outcome Variables

The Japanese version of FoCS was used to evaluate fear and anxiety related to COVID-19 [9,28,29]. Its validity and reliability were confirmed. The FoCS consists of the following seven items: “I am most afraid of COVID-19”, “It makes me uncomfortable to think about COVID-19”, “My hands become clammy when I think about COVID-19”, “I am afraid of losing my life because of COVID-19”, “When watching news and stories about COVID-19 on social media, I become nervous or anxious”, “I cannot sleep because I am worried about getting COVID-19”, and “My heart races or palpitates when I think about getting COVID-19.” The subjects responded to each item on a five-point option ranging from “strongly disagree (score of 0)” to “strongly agree (score of 5)”. The FoCS scores were analyzed both as a continuous variable and as a categorical variable. For the latter purpose, the subjects were dichotomized at the median, 18, of the total score.

The question “How likely do you think you will be infected with COVID-19?” was used to assess perceived susceptibility [15]. They chose one of the five options: 1. Certainly not (0%), 2. Possibly (25%), 3. Maybe (50%), 4. Probably (75%), and 5. Certainly (100%). Respondents who chose 4 or 5 were considered perceiving susceptibility to COVID-19.

### 2.4. Potential Confounding Factors

We considered the following factors as potential confounding factors: gender, age, having a spouse, comorbidities (NCDs and depression), time spent watching news programs broadcasted by commercial TV stations, employment status, and household income [2,7,30,31,32,33,34]. Ages were categorized as 15–19, 20–29, 30–39, 40–49, 50–59, 60–69, and 70–80 years old. Spouses were defined as a wife/husband or a person who was living together as a de facto couple but had not officially been registered as a married couple. Time spent watching news programs broadcasted by commercial TV stations was classified as follows: none, once to 3 times per month, 1 day per week, 2 to 3 days per week, 4 to 5 days per week, and almost every day. In this study, NCDs included hypertension, diabetes, asthma, pneumonia/bronchitis, angina pectoris, myocardial infarction, stroke (cerebral infarction or hemorrhage), chronic obstructive pulmonary disease, chronic kidney disease, chronic liver disease (excluding fatty liver or chronic hepatitis), immune deficiency disease (including those under steroid administration), and cancer. Regarding employment status, respondents were asked whether they were regularly working (self-employed, employed), not regularly working (dispatched/contract/outsourced worker, part-timer, in-house worker), or not working (student, retired, housewife/househusband, unemployed). Household income was categorized as less than 3, 3 or more and less than 5, 5 or more and less than 8, 8 or more and less than 10, 10 or more million Japanese Yen (JPY), and missing response. As of 14 November 2022, 100 JPY was equivalent to 0.72 US dollars.

### 2.5. Statistical Analysis

The characteristics of the subjects were compared by gender using t-test and chi-square test. We examined whether gender and the potential confounding factors were associated with the FoCS score and perceived susceptibility to COVID-19 using t-test, analysis of variance, and chi-square test. Effect sizes were calculated. The standards of effect sizes are introduced in Appendix A [35,36,37]. Using multiple linear regression analysis, standardized betas of gender and the potential confounding factors for FoCS score were calculated. Using multiple logistic regression analysis, we calculated the odds ratios of gender for having a FoCS score above the median and perceiving susceptibility to COVID-19. The potential confounding factors were adjusted for calculating the standardized betas and odds ratios. In the multivariable analyses, age was used as a continuous variable for avoiding multicollinearity. As sensitivity analyses, statistical analyses were also performed for all respondents (*n* = 28,000). Statistical significance was set at *p* < 0.05. Analytic calculations were performed using IBM SPSS 24.0.

## 3. Results

The characteristics of the subjects are shown in Table 1. Women had a higher mean FoCS score than men (18.6 vs. 17.5) and a higher proportion of having the median or higher score (57.4% vs. 51.4%) than men. They also perceived susceptibility to COVID-19 more frequently than men (13.6% vs. 11.5%). Age was higher in men than in women, but the difference was slight. The prevalence of comorbidities was higher in men than women. Men had a spouse more frequently than women. Women were more likely to watch commercial TV stations’ news programs, irregularly or never work, and have low household income than men. Women gave missing responses regarding household income more frequently than men.

Women significantly had higher FoCS scores than men, although the effect size was small. Table 2 indicates the mean (standard deviation: SD) FoCS score by gender and the potential confounding factor. All the variables were significantly associated with the FoCS scores. When using multiple linear regression analysis, the association between gender and the FoCS scores remained significant (Table 3). Having NCDs, depression, and a spouse, frequently watching commercial TV stations’ news programs, and household income were significantly associated with the FoCS scores, while age and employment status were not.

Women reported a median or higher FoCS score more frequently than men (Table 4). The effect size was small. All the variables were significantly associated with reporting a median or higher FoCS score. Even in multiple logistic regression analysis, gender was significantly associated with reporting a median or higher FoCS score (Table 5). Having NCDs, depression, and a spouse, and a low frequency of watching commercial TV stations’ news programs were associated with reporting a median or higher FoCS score. High household income had a low probability of reporting a median or higher FoCS score. High frequency of watching commercial TV stations’ news programs and employment status were not associated with reporting a median or higher FoCS score.

The proportions of and odds ratios for perceived susceptibility to COVID-19 are shown in Table 6 and Table 7, respectively. Women perceived the susceptibility to COVID-19 approximately 1.3-fold more frequently than men. The statistical significance remained even after adjustment for the potential confounding factors. Having NCDs and depression and a low frequency of watching commercial TV stations’ news programs were associated with perceived susceptibility to COVID-19. Not working regularly had a low probability of perceived susceptibility to COVID-19. We failed to find an association between household income classes and perceived susceptibility prominent.

Sensitivity analyses using all respondents resulted in similar findings that women had higher FoCS scores and perceived susceptibility to COVID-19 than men (Appendix A).

## 4. Discussion

We found in the present study that women had higher FoCS scores than men even one year after the pandemic occurred in Japan. Women were more likely than men, approximately 1.3-fold higher, to feel fear and anxiety about COVID-19 when the FoCS scores were divided at the median score. Women perceived the susceptibility to COVID-19 approximately 1.3-fold higher than men.

An advantage of the present study is showing that the gender differences in the FoCS scores and perceived susceptibility to COVID-19 remained significant even after adjustment for age, having comorbidities and a spouse, time spent watching news programs broadcasted by commercial TV stations, employment status, and household income. Gender differences do not merely reflect biological differences. They might be a reflection of the differences in socioeconomic and health statuses and lifestyles by gender. In the present study, women and men showed different proportions of having comorbidities and a spouse, time spent watching news programs broadcasted by commercial TV stations, employment status, and household income. A large sample size of the present study allowed the adjustment of these factors.

Our findings regarding fear and anxiety were concordant with those from other countries [1,2,3,4,7,8,10]. We observed in this study that women were more anxious and fearful about infectious diseases during a pandemic, even approximately one year after the onset of the COVID-19 pandemic in Japan. Women were reportedly more fearful and anxious about infectious diseases during previous infectious disease pandemics. Hong Kong experienced the SARS pandemic from 2003 to 2004. Leung et al. [12] reported that women felt more anxiety in the early stages of the SARS pandemic than men, while the difference disappeared in the late stage of the pandemic. Lau et al. [11] reported that women were more likely to report mental illness and feel fearful about SARS than men. A study conducted during the 2018 MERS outbreak in South Korea found that women health care workers felt psychological distress more frequently than male health care workers [13].

Here, we discuss the trend of the FoCS scores of general adults in Japan. The first patient infected with COVID-19 was found on 15 January 2020 in Japan. Midorikawa et al. [9] addressed a general adult population in Japan in early August 2020, i.e., amid the second wave of the COVID-19 pandemic. They reported a mean (SD) FoCS score of 15.3 (4.9) for men, 17.4 (4.6) for women, and 17.3 (5.8) for others. There was a significant difference by gender. The effect size was small to the medium: η^2^ = 0.042. Our study was conducted in February 2021, i.e., six months later of their study. We found a mean (SD) FoCS score of 17.5 (5.9) for men and 18.6 (5.6) for women. The effect size was small: d = 0.190. The subjects of the two studies were not identical. Therefore, strictly saying, it may be inappropriate to directly compare our findings to theirs. However, both men and women showed higher mean FoCS scores and the difference in the mean FoCS by gender became smaller in our study than in Midorikawa’s study [9].

In our study, women perceived susceptibility to COVID-19 more frequently than men. This finding is concordant with the previous findings regarding COVID-19 [14,15] and other infectious disease pandemics, such as SARS [11,18] and avian influenza [19]. Conversely, some previous studies found no difference between men and women in the perceived susceptibility to COVID-19 [16,17] and influenza A virus subtype H5N1 in Asia [20]. Further studies are necessary to determine the gender difference in the perceived susceptibility to COVID-19 and other infectious disease pandemics.

Regarding the associations with the FoCS scores and perceived susceptibility to COVID-19, gender showed a low effect size. We consider having NCDs, depression, and a spouse, exposure to media, employment status, and household income in the present study since they were reported to have potential associations with fear and anxiety about and perceived susceptibility to COVID-19 [2,7,30,31,32,33,34]. We confirmed in this study that some of them were associated with the FoCS scores and perceived susceptibility to COVID-19. At the same time, we found their effect sizes small. At least, we could not conclude that gender was less important than those factors in terms of the impact on fear and anxiety about and perceived susceptibility to COVID-19.

We here discuss age, having a spouse, and employment status (not working) whose associations with FoCS scores were different from those with perceived susceptibility to COVID-19. It was not clearly shown that FoCS scores increased with age. Having a spouse was significantly associated with high FoCS scores even after adjustment for age and other potential confounding factors. On the other hand, perceived susceptibility to COVID-19 was inversely associated with age. A similar finding was observed in a study in Israel [17]. The association between having a spouse and perceived susceptibility to COVID-19 was not significant in the multivariable analysis. This would suggest that having a spouse was a confounding factor for the association between age and perceived susceptibility to COVID-19. In our study, the prevalence of having a spouse was higher in the older age than in the younger age (Appendix A). Not working was not associated with FoCS scores but with perceived susceptibility to COVID-19. Compared to those who were working, those who were not working would have fewer opportunities to meet other people outside, leading to less perceived susceptibility to COVID-19.

The present study had some limitations. First, there might be a selection bias that the subjects were limited to those interested in this kind of internet survey. We could not calculate the response rate in this study. This was because the enrollment of subjects closed when the respondents reached the target number that was set to reflect the gender, age, and geographical distribution of the Japanese general population. The subjects might be limited to those who had an interest in this survey and could go through a large number of questions, which is called the volunteer effect [38]. In addition, the availability of the internet differed by age and household income, which could limit the representativeness of the subjects. According to the latest white paper on information and communications in Japan [39], those in their sixties (82.7%) and seventies (59.6%) were using the internet less frequently than younger adults (more than 95%), and household income was positively correlated with the availability of the internet. The distribution of the household income of the present subjects did not appear different from the latest national survey finding [40], although it must be noted that a considerable proportion of the present subjects, especially women, did not disclose their household income. Second, we did not collect data on nationality, ethnicity, or religious affiliation. Such information might have contributed to the discussion regarding the impact of cultural differences on the gender difference in fear and anxiety about and perceived susceptibility to COVID-19.

## 5. Conclusions

In this cross-sectional study, we clarified that women felt fear and anxiety about and perceived susceptibility to COVID-19 more frequently than men, using the data collected through a web-based survey during the third wave of the COVID-19 pandemic in Japan.

## Figures and Tables

**Table 1 ijerph-19-16239-t001:** Characteristics of subjects.

	Men (*n* = 11,957)	Women (*n* = 11,559)	*p* Value
Age	50.5 (16.4)	49.8 (17.1)	0.001
15–19	234 (2.0%)	385 (3.3%)	<0.001
20–29	1355 (11.3%)	1467 (12.7%)	
30–39	1687 (14.1%)	1559 (13.5%)	
40–49	2404 (20.1%)	2183 (18.9%)	
50–59	2229 (18.6%)	2048 (17.7%)	
60–69	2187 (18.3%)	2127 (18.4%)	
70–80	1861 (15.6%)	1790 (15.5%)	
Score of the Fear of COVID-19 Scale	17.5 (5.9)	18.6 (5.6)	<0.001
Having a median or higher score	6145 (51.4%)	6635 (57.4%)	<0.001
Perceived susceptibility to COVID-19	1370 (11.5%)	1569 (13.6%)	<0.001
Comorbidity			
Non-communicable diseases	3932 (32.9%)	2336 (20.2%)	<0.001
Depression	497 (4.2%)	399 (3.5%)	0.005
Having a spouse	7454 (62.3%)	6815 (59.0%)	<0.001
Frequency of watching commercial TV stations’ news programs			
None	2357 (19.7%)	1999 (17.3%)	<0.001
Once to 3 times/month	929 (7.8%)	691 (6.0%)	
1 day/week	927 (7.8%)	718 (6.2%)	
2 to 3 days/week	1452 (12.1%)	1281 (11.1%)	
4 to 5 days/week	1520 (12.7%)	1470 (12.7%)	
Almost every day	4772 (39.9%)	5400 (46.7%)	
Employment status			
Regularly working	7377 (61.7%)	3053 (26.4%)	<0.001
Not regularly working	1327 (11.1%)	2816 (24.4%)	
Not working	3253 (27.2%)	5690 (49.2%)	
Household income (Japanese Yen)			
<3 million	1923 (16.1%)	2247 (19.4%)	<0.001
3 million or more and less than 5 million	2699 (22.6%)	2476 (21.4%)	
5 million or more and less than 8 million	2834 (23.7%)	2222 (19.2%)	
8 million or more and less than 10 million	1219 (10.2%)	770 (6.7%)	
10 million or more	1442 (12.1%)	906 (7.8%)	
Missing response	1840 (15.4%)	2938 (25.4%)	

Figures are presented as the mean (standard deviation) or the number (proportion). *p* values were calculated with *t*-test or chi-square test.

**Table 2 ijerph-19-16239-t002:** Mean (standard deviation: SD) of Fear of COVID-19 Scale score.

	Mean (SD)	*p* Value	Effect Size
Gender			
Men	17.5 (5.9)	<0.001	0.190
Women	18.6 (5.6)		
Age			
15–19	18.6 (6.1)	<0.001	0.004
20–29	17.8 (6.2)		
30–39	17.6 (6.1)		
40–49	17.8 (6.0)		
50–59	18.0 (5.6)		
60–69	18.4 (5.4)		
70–80	18.7 (5.3)		
Comorbidity			
Non-communicable diseases			
Absent	17.9 (5.9)	<0.001	0.121
Present	18.6 (5.5)		
Depression			
Absent	18.0 (5.7)	<0.001	0.253
Present	19.5 (6.4)		
Having a spouse			
Not applicable	17.8 (5.9)	<0.001	0.073
Applicable	18.2 (5.7)		
Frequency of watching commercial TV stations’ news programs			
None	17.4 (6.7)	<0.001	0.003
Once to 3 times/month	18.4 (6.2)		
1 day/week	18.4 (5.7)		
2 to 3 days/week	18.1 (5.6)		
4 to 5 days/week	18.1 (5.4)		
Almost every day	18.3 (5.4)		
Employment status			
Regularly working	17.6 (6.0)	<0.001	0.005
Not regularly working	18.3 (5.7)		
Not working	18.5 (5.6)		
Household income (Japanese Yen)			
<3 million	18.3 (5.9)	<0.001	0.007
3 million or more and less than 5 million	18.2 (5.6)		
5 million or more and less than 8 million	17.7 (5.6)		
8 million or more and less than 10 million	17.5 (5.7)		
10 million or more	17.2 (5.8)		
Missing response	18.8 (5.8)		

Effect size is expressed by Cohen’s d (gender, comorbidity, having a spouse) or η^2^ (age, frequency of watching news, employment status, and household income).

**Table 3 ijerph-19-16239-t003:** Standardized betas for Fear of COVID-19 Scale scores: multiple linear regression analysis.

	Standardized Beta
Gender	
Men	Reference
Women	0.092 ***
Age	0.012
Comorbidity	
Non-communicable diseases	
Absent	Reference
Present	0.050 ***
Depression	
Absent	Reference
Present	0.048 ***
Having a spouse	
Not applicable	Reference
Applicable	0.045 ***
Frequency of watching commercial TV stations’ news programs	
None	Reference
Once to 3 times/month	0.046 ***
1 day/week	0.046 ***
2 to 3 days/week	0.041 ***
4 to 5 days/week	0.035 ***
Almost every day	0.051 ***
Employment status	
Regularly working	Reference
Not regularly working	−0.002
Not working	0.001
Household income (Japanese Yen)	
<3 million	Reference
3 million or more and less than 5 million	−0.018 *
5 million or more and less than 8 million	−0.045 ***
8 million or more and less than 10 million	−0.039 ***
10 million or more	−0.058 ***
Missing response	0.025 ***

All independent variables were included in the multiple linear regression model for calculating adjusted odds ratios. *: *p* < 0.05 ***: *p* < 0.001.

**Table 4 ijerph-19-16239-t004:** Proportions of reporting a median or higher Fear of COVID-19 Scale score.

	*n* (%)	*p* Value	Effect Size
Gender			
Men	6145 (51.4%)	<0.001	0.060
Women	6635 (57.4%)		
Age			
15–19	366 (59.1%)	<0.001	0.053
20–29	1507 (53.4%)		
30–39	1645 (50.7%)		
40–49	2426 (52.9%)		
50–59	2263 (52.9%)		
60–69	2428 (56.3%)		
70–80	2145 (58.8%)		
Comorbidity			
Non-communicable diseases			
Absent	9170 (53.2%)	<0.001	0.039
Present	3610 (57.6%)		
Depression			
Absent	12208 (54.0%)	<0.001	0.038
Present	572 (63.8%)		
Having a spouse			
Not applicable	4921 (53.2%)	0.005	0.018
Applicable	7859 (55.1%)		
Frequency of watching commercial TV stations’ news programs			
None	2262 (51.9%)	0.006	0.026
Once to 3 times/month	910 (56.2%)		
1 day/week	924 (56.2%)		
2 to 3 days/week	1494 (54.7%)		
4 to 5 days/week	1608 (53.8%)		
Almost every day	5582 (54.9%)		
Employment status			
Regularly working	5343 (51.2%)	<0.001	0.056
Not regularly working	2339 (56.5%)		
Not working	5098 (57.0%)		
Household income (Japanese Yen)			
<3 million	2366 (56.7%)	<0.001	0.084
3 million or more and less than 5 million	2843 (54.9%)		
5 million or more and less than 8 million	2588 (51.2%)		
8 million or more and less than 10 million	974 (49.0%)		
10 million or more	1122 (47.8%)		
Missing response	2887 (60.4%)		

Effect size is expressed by φ (gender, comorbidity, having a spouse) or Cramer’s V (age, frequency of watching news, employment status, and household income).

**Table 5 ijerph-19-16239-t005:** Odds ratios for reporting a median or higher Fear of COVID-19 Scale score: multiple logistic regression analysis.

	Odds Ratio (95% Confidence Interval)	
	Crude	Adjusted ^1^
Gender		
Men	Reference	Reference
Women	1.27 (1.21–1.34) ***	1.24 (1.17–1.32) ***
Age	1.00 (1.00–1.00) ***	1.00 (1.00–1.00)
Comorbidity		
Non-communicable diseases		
Absent	Reference	Reference
Present	1.20 (1.13–1.27) ***	1.18 (1.11–1.26) ***
Depression		
Absent	Reference	Reference
Present	1.51 (1.31–1.73) ***	1.49 (1.29–1.71) ***
Having a spouse		
Not applicable	Reference	Reference
Applicable	1.08 (1.02–1.14) **	1.14 (1.07–1.21) ***
Frequency of watching commercial TV stations’ news programs		
None	Reference	Reference
Once to 3 times/month	1.19 (1.06–1.33) **	1.22 (1.09–1.37) ***
1 day/week	1.19 (1.06–1.33) **	1.23 (1.09–1.38) ***
2 to 3 days/week	1.12 (1.01–1.23) *	1.13 (1.03–1.25) *
4 to 5 days/week	1.08 (0.98–1.18)	1.06 (0.97–1.17)
Almost every day	1.13 (1.05–1.21) **	1.06 (0.98–1.14)
Employment status		
Regularly working	Reference	Reference
Not regularly working	1.23 (1.15–1.33) ***	1.04 (0.96–1.12)
Not working	1.26 (1.19–1.34) ***	1.03 (0.96–1.10)
Household income (Japanese Yen)		
<3 million	Reference	Reference
3 million or more and less than 5 million	0.93 (0.86–1.01)	0.92 (0.85–1.00)
5 million or more and less than 8 million	0.80 (0.74–0.87) ***	0.80 (0.73–0.88) ***
8 million or more and less than 10 million	0.73 (0.66–0.81) ***	0.74 (0.66–0.83) ***
10 million or more	0.70 (0.63–0.77) ***	0.70 (0.63–0.79) ***
Missing response	1.16 (1.07–1.27) ***	1.15 (1.06–1.26) **

All independent variables were included in the multiple logistic regression model included for calculating adjusted odds ratios. *: *p* < 0.05; **: *p* < 0.001; ***: *p* < 0.001.

**Table 6 ijerph-19-16239-t006:** Proportions of perceived susceptibility to COVID-19.

	*n* (%)	*p*-Value	Effect Size
Gender			
Men	1370 (11.5%)	<0.001	0.032
Women	1569 (13.6%)		
Age			
15–19	123 (19.9%)	<0.001	0.081
20–29	495 (17.5%)		
30–39	457 (14.1%)		
40–49	584 (12.7%)		
50–59	478 (11.2%)		
60–69	434 (10.1%)		
70–80	368 (10.1%)		
Comorbidity			
Non-communicable diseases			
Absent	2126 (12.3%)	0.186	0.009
Present	813 (13.0%)		
Depression			
Absent	2752 (12.2%)	<0.001	0.050
Present	187 (20.9%)		
Having a spouse			
Not applicable	1302 (14.1%)	<0.001	0.039
Applicable	1637 (11.5%)		
Frequency of watching commercial TV stations’ news programs			
None	569 (13.1%)	<0.001	0.042
Once to 3 times/month	254 (15.7%)		
1 day/week	252 (15.3%)		
2 to 3 days/week	349 (12.8%)		
4 to 5 days/week	367 (12.3%)		
Almost every day	1148 (11.3%)		
Employment status			
Regularly working	1330 (12.8%)	0.008	0.020
Not regularly working	561 (13.5%)		
Not working	1048 (11.7%)		
Household income (Japanese Yen)			
<3 million	565 (13.5%)	0.027	0.023
3 million or more and less than 5 million	636 (12.3%)		
5 million or more and less than 8 million	592 (11.7%)		
8 million or more and less than 10 million	250 (12.6%)		
10 million or more	265 (11.3%)		
Missing response	631 (13.2%)		

Effect size is expressed by φ (gender, comorbidity, having a spouse) or Cramer’s V (age, frequency of watching news, employment status, and household income).

**Table 7 ijerph-19-16239-t007:** Odds ratios for perceived susceptibility to COVID-19: multiple logistic regression analysis.

	Odds Ratio (95% Confidence Interval)	
	Crude	Adjusted ^1^
Gender		
Men	Reference	Reference
Women	1.21 (1.12–1.31) ***	1.27 (1.16–1.38) ***
Age	0.99 (0.98–0.99) ***	0.99 (0.98–0.99) ***
Comorbidity		
Non-communicable diseases		
Absent	Reference	Reference
Present	1.06 (0.97–1.16)	1.34 (1.22–1.48) ***
Depression		
Absent	Reference	Reference
Present	1.90 (1.61–2.24) ***	1.63 (1.37–1.93) ***
Having a spouse		
Not applicable	Reference	Reference
Applicable	0.79 (0.73–0.85) ***	1.00 (0.91–1.10)
Frequency of watching commercial TV stations’ news programs		
None	Reference	Reference
Once to 3 times/month	1.24 (1.05–1.45) **	1.27 (1.08–1.49) **
1 day/week	1.20 (1.03–1.41) *	1.24 (1.06–1.46) **
2 to 3 days/week	0.97 (0.84–1.12)	1.05 (0.90–1.21)
4 to 5 days/week	0.93 (0.81–1.07)	1.04 (0.90–1.20)
Almost every day	0.85 (0.76–0.94) **	1.00 (0.89–1.12)
Employment status		
Regularly working	Reference	Reference
Not regularly working	1.07 (0.96–1.19)	1.00 (0.89–1.12)
Not working	0.91 (0.83–0.99) *	0.89 (0.81–0.99) *
Household income (Japanese Yen)		
<3 million	Reference	Reference
3 million or more and less than 5 million	0.89 (0.79–1.01)	0.90 (0.79–1.02)
5 million or more and less than 8 million	0.85 (0.75–0.96) **	0.81 (0.71–0.92) **
8 million or more and less than 10 million	0.92 (0.78–1.08)	0.88 (0.74–1.04)
10 million or more	0.81 (0.69–0.95) **	0.78 (0.66–0.93) **
Missing response	0.97 (0.86–1.10)	0.94 (0.82–1.06)

All independent variables were included in the multiple logistic regression model for calculating adjusted odds ratios. *: *p* < 0.05; **: *p* < 0.001; ***: *p* < 0.001.

## Data Availability

The data used in this study are not available in a public repository because they contain personally identifiable or potentially sensitive patient information. Based on the regulations for ethical guidelines in Japan, the Research Ethics Committee of the Osaka International Cancer Institute has imposed restrictions on the dissemination of the data collected in this study. All data inquiries should be addressed to the person responsible for data management, Takahiro Tabuchi at the following e-mail address: tabuchitak@gmail.com.

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
