# Peer review of "Gender Difference in Fear and Anxiety about and Perceived Susceptibility to COVID-19 in the Third Wave of Pandemic among the Japanese General Population: A Nationwide Web-Based Cross-Sectional Survey"

_ijerph, 2022, doi:10.3390/ijerph192316239_

Round 1

Reviewer 1 Report

The manuscript concerns a cross-sectional, internet-based questionnaire on COVID-19 fear/anxiety and susceptibility. The paper is small in scope, addressing two straightforward and highly-related questions. With the limitations inherent to internet-based research (e.g. low response rates, though conversely a very large sample size) in mind, I think this paper contributes to achieving a more world-wide (rather than exclusively Western, as is too often the case) view of gender differences in COVID-19 fear/anxiety and susceptibility. I have two main points for improvement:

First, as mentioned, the paper is very small in scope. The authors have the data to easily make the paper more substantial by comparing the importance of different factors (e.g. gender, age, chronic illness, mental health issues, exposure to news, etc.). I think this would substantially increase the contribution of the manuscript to the literature.

Second, the statistical analysis is based on dichotomizing outcomes, which is not recommended. I would advise the authors to perform simple linear regression on the continuous FoCS scale and ordinal logistic regression on the ordinal perceived susceptibility question. The authors’ sample size is large enough that power loss is not much of an issue, but dichotomization is a suboptimal practice that should be avoided whenever possible.

Smaller points:

Introduction

1.     “Few studies examined whether women felt fear or anxiety about COVID-19 more frequently than men in Japan” – please provide references to these few studies or state that there are none (to the authors’ knowledge) rather than few.

2.     “Women were reported to perceive susceptibility to COVID-19” presumably means “Women were reported to perceive greater susceptibility to COVID-19 than men

Methods

1.     The target sample of 28,000 participants was not reached, even though the method states that “enrolment continued until the target number of respondents was reached”. Why is that? Please also state the number of participants who were excluded from the survey because they did not answer “second from last” to the attention check question. If this is a substantial number, it may be worthwhile to perform sensitivity analyses with and without these participants.

2.     The answer options to the question “How likely do you think you will be infected with COVID-19?” seem suboptimal (“not applicable at all” to “very applicable” does not clearly relate to “how likely”). Possibly this is related to the translation from Japanese to English, but otherwise it seems like this may have affected participants’ answers.

3.     I can see the logic behind categorizing age, as it seems likely that there is a true qualitative difference between older people and younger people, but I would use more than 2 categories. It might also be worthwhile to explicitly investigate whether the association between age and the outcome measure appears to be nonlinear.

Results

1.     Please provide full model results (i.e. including odds ratios for confounders).

Discussion

1.     The difference between men and women in fear of COVID-19 and perceived susceptibility is very highly significant due to the large sample size, but small in terms of effect size (SMD of about 1/6 or 0.17 for the FoCS scale). The odds ratios in the tables (OR=1.2-1.3) are also generally considered pretty small. This is fine, but I would expect the authors to reflect on the effect size in the discussion and not just on its statistical significance. Were gender differences in other studies also quite small or more substantial? Is this really an important gender difference to take into account for policy-making etc. or might other factors be more relevant (e.g. age, chronic illness, general vulnerability to COVID-19)?

2.     In general, the discussion is somewhat superficial and there is not a lot of critical engagement with the authors’ own results or with other researchers’ results or e.g. with the possibility of cultural differences which presumably motivates the paper.

Author Response

Reply to the review comments

We appreciate the editor for the opportunity to submit the revised manuscript. We also appreciate the reviewers’ precious comments. They help us improve our manuscript. Below is our reply to the review comments. We highlighted our revision in yellow in our revised manuscript.

Correction

We found a mistake in the data source during revising the manuscript. Our data do not come from the Japan COVID-19 and Society and New Tobacco Internet Survey (JACSIS) that was conducted between August and September 2020. The data come from the Japan Society and New Tobacco Internet Survey that was conducted in February 2021(JASTIS 2021). It was in the third wave of the pandemic and approximately one year after the first patient in Japan. The numbers of subjects were different between the two surveys: n = 26,000 for the JASTIS 2021 and n = 28,000 for the JACSIS. When the survey was conducted was different between the two surveys. We have accordingly revised the discussion so that we discuss whether gender affected the fear and anxiety about and perceived susceptibility to COVID-19 even nearly one year after the onset of the pandemic.

Revision

Materials and Methods, lines 62 – 85

Reviewer 1

  • The manuscript concerns a cross-sectional, internet-based questionnaire on COVID-19 fear/anxiety and susceptibility. The paper is small in scope, addressing two straightforward and highly-related questions. With the limitations inherent to internet-based research (e.g. low response rates, though conversely a very large sample size) in mind, I think this paper contributes to achieving a more world-wide (rather than exclusively Western, as is too often the case) view of gender differences in COVID-19 fear/anxiety and susceptibility. I have two main points for improvement:

Response

We appreciate your high evaluation of our paper. Following your comment, we have revised the discussion on the possible selection biases related to internet-based surveys. We cannot calculate the response rate in this study because the sample enrollment closed when the participants reached the target number.

Revision

Discussion, lines 296 - 310

  • First, as mentioned, the paper is very small in scope. The authors have the data to easily make the paper more substantial by comparing the importance of different factors (e.g. gender, age, chronic illness, mental health issues, exposure to news, etc.). I think this would substantially increase the contribution of the manuscript to the literature.

Response

In our revised manuscript, we have shown the associations of all the independent variables (age, comorbidity, spouse, exposure to news, employment status, and household income) with the Fear of COVID-19 Scale (FoCS) score and the susceptibility to COVID-19. This enabled the discussion on whether gender had affected the FoCS score and the susceptibility independent of and as well as other factors.

Revision

Results, lines 158 – 165, 202 – 210, and 224 - 228

New Tables 2 – 7 have been made.

Discussion, lines 247 – 256 and 287 - 295

  • Second, the statistical analysis is based on dichotomizing outcomes, which is not recommended. I would advise the authors to perform simple linear regression on the continuous FoCS scale and ordinal logistic regression on the ordinal perceived susceptibility question. The authors’ sample size is large enough that power loss is not much of an issue, but dichotomization is a suboptimal practice that should be avoided whenever possible.

Response

In the revised manuscript, we have treated the FoCs score both as a continuous variable and as a dichotomized variable (whether to have a median or higher score or not). In both cases, we have found a significant association between gender and the FoCS scores.

We have also classified age and exposure to news into more categories. In the multivariable analyses, age was included as a continuous variable to avoid multicollinearity.

Revision

Materials and Methods (2.3. Outcome Variables), lines 101 – 103, 114 – 117, and 125 – 128

Materials and Methods (2.5. Statistical Analysis), lines 134 - 140

New Tables 2 – 7 have been made.

Smaller points:

Introduction

  1. “Few studies examined whether women felt fear or anxiety about COVID-19 more frequently than men in Japan” – please provide references to these few studies or state that there are none (to the authors’ knowledge) rather than few.

Response

Midorikawa et al. examined the FoCS scores of Japanese adults in August 2020. We have mentioned it in the introduction. To our knowledge, no studies have addressed perceived susceptibility to COVID-19.

Revision

Introduction, lines 37 – 38 and 49 - 50

  1. “Women were reported to perceive susceptibility to COVID-19” presumably means “Women were reported to perceive greatersusceptibility to COVID-19 than men

 Response

We have revised the sentence, following your comment.

Revision

Introduction, lines 44 - 45

Methods

  1. The target sample of 28,000 participants was not reached, even though the method states that “enrolment continued until the target number of respondents was reached”. Why is that? Please also state the number of participants who were excluded from the survey because they did not answer “second from last” to the attention check question. If this is a substantial number, it may be worthwhile to perform sensitivity analyses with and without these participants.

Response

The correct target number of the sample was 26,000, not 28,000, for the JASTIS 2021 Study. We got a total of 26,000 respondents in the present study. Of them, 2484 (9.6%) did not correctly answer the item “Choose the penultimate option from the following five options” and were excluded from the analyses.

However, following your comment, we have analyzed all of the 26,000 respondents as sensitivity analyses. They resulted in similar findings: women significantly had a higher mean FoCS score, indicated a higher proportion of having the median or higher score, and perceived the perceived susceptibility to COVID-19 more frequently than men (Supplementary Tables 2 – 7).

Revision

Materials and Methods (2.2. Subjects), lines 87 – 92

Materials and Methods (2.5. Statistical Analysis), lines 141 – 142

Results, lines 238 – 240

We have made new Supplementary Tables 2 – 7.

  1. The answer options to the question “How likely do you think you will be infected with COVID-19?” seem suboptimal (“not applicable at all” to “very applicable” does not clearly relate to “how likely”). Possibly this is related to the translation from Japanese to English, but otherwise it seems like this may have affected participants’ answers.

Response

We have revised the translation as follows: 1. Certainly not (0%), 2. Possibly (25%), 3. Maybe (50%), 4. Probably (75%), and 5. Certainly (100%).

Revision

Materials and Methods (2.3 Outcome Variables), lines 106 – 107

  1. I can see the logic behind categorizing age, as it seems likely that there is a true qualitative difference between older people and younger people, but I would use more than 2 categories. It might also be worthwhile to explicitly investigate whether the association between age and the outcome measure appears to be nonlinear.

Response

Age has been categorized basically in the 10-year age range (15 – 19, 20 – 29, 30 – 39, 40 – 49, 50 – 59, 60 – 69, and 70 – 80 years old). We have also made exposure to news categorized into more classes.

In response to Reviewer 2’s comment, we have added yearly household income as an independent variable. Yearly household income was classified into six categories.

Revision

Materials and Methods (2.4 Potential Confounding Factors), lines 110 – 114, 115 – 118, and 126 – 129

New Tables 1 – 7 have been made.

Results

  1. Please provide full model results (i.e. including odds ratios for confounders).

Response

We have shown the full model results, following your suggestion.

Revision

Results, lines 160 – 165, 203 – 210, and 224 - 228

New Tables 2 – 7 have been made.

Discussion

  1. The difference between men and women in fear of COVID-19 and perceived susceptibility is very highly significant due to the large sample size, but small in terms of effect size (SMD of about 1/6 or 0.17 for the FoCS scale). The odds ratios in the tables (OR=1.2-1.3) are also generally considered pretty small. This is fine, but I would expect the authors to reflect on the effect size in the discussion and not just on its statistical significance. Were gender differences in other studies also quite small or more substantial? Is this really an important gender difference to take into account for policy-making etc. or might other factors be more relevant (e.g. age, chronic illness, general vulnerability to COVID-19)?

Response

We have calculated the effect size not only for gender but also for the other independent variables. This has enabled us to discuss how much gender affected fear and anxiety about and perceived susceptibility to COVID-19.

Revision

Materials and Methods (2.5 Statistical Analysis), lines 134 – 135

We have newly made Tables 2, 4, and 6.

Discussion, lines 287 - 294

  1. In general, the discussion is somewhat superficial and there is not a lot of critical engagement with the authors’ own results or with other researchers’ results or e.g. with the possibility of cultural differences which presumably motivates the paper.

Response

Reviewer 2 also found our discussion superficial. We have improved it as follows:

  • 2nd paragraph: We have shown the advantage of our study that we considered socioeconomic and health statuses and lifestyles that differed by gender in the analyses.
  • 4th paragraph: We have discussed whether the FoCS scores of Japanese general people changed as time went by.
  • 6th paragraph: We have discussed the effect size of age and the potential confounding factors on the FoCS scores and perceived susceptibility to COVID-19.
  • 7th paragraph: We have discussed the study limitations in more detail. One topic is the potential selection biases related to the internet-based survey. Another topic concerns the absence of data on nationality, ethnicity, and religious affiliation. This made us difficult to discuss the possibility of cultural differences.

Revision

Discussion, lines 247 – 256, 268 – 279, 287 – 295, and 296 - 313

Reviewer 2 Report

Abstract.

Authors stated: “It may be universal that women are more anxious and fearful about and perceived susceptibility to infectious diseases”.

I think it is a difficult statement to support . and I also think it is widely disputable to be able to define something potentially universal with such a shallow study . Remaining in doubt as to what might be universal .

Introduction.

The introduction gives some information on existing studies with respect to gender differences in relation to fear of contagion - but does not discuss any other characteristics.  Women are mentioned as a uniform category, failing to get any information - due in an introduction - with respect to other variables(socio-economic status, ethnicity..etc – intersectionality)

They say in the introduction that the results can help develop interventions and health policies. On this introduction? Health policy only relative to the issue of being in the woman category? Not including an intersectional lens is a bit anachronistic nowadays. There are many studies, for example done in countries where there is a great deal of discrimination related to class, ethnicity .. (see US or Israel), that have highlighted how these variables meet together in increasing perceived risk/actual risk of infection.

The authors then include it in the confounding factors (where they leave out nationality, ethnicity, or religious affiliation..) but do not mention it in the introduction, - nor elsewhere in the text, which would be needed.

Method and participants.

Difficult to express with the fact that the authors say the details are described elsewhere.

Dividing the subjects into two bands, 15-64 and 65-49 is a bit reductive. Wasn't it possible to have a few more bands?

Does having job information with the required categories (regular, not regular) give us any indication with respect to (in)sufficient income? I guess this could have a major implication in any person concerns about Covid or any other disease. In the US, for instance - being able to have an high salary or not have a major influence on my access to health facilities.. and not just in the US..

The discussion, like the rest of the work in my opinion is very superficial.

A possible explanation is that women tend to be more concerned about their own and their family’s health and seek medical care more frequently than men”

Again, it seems to me that there is an enormity of literature-to date and even in the last 20 years-that exposes how this is not an explanation... What variables come into play for women to be more concerned-or for men to be less concerned-about health ? it seems to me that this study doesn't tell us much...

Author Response

Reply to the review comments

We appreciate the editor for the opportunity to submit the revised manuscript. We also appreciate the reviewers’ precious comments. They help us improve our manuscript. Below is our reply to the review comments. We highlighted our revision in yellow in our revised manuscript.

Correction

We found a mistake in the data source during revising the manuscript. Our data do not come from the Japan COVID-19 and Society and New Tobacco Internet Survey (JACSIS) that was conducted between August and September 2020. The data come from the Japan Society and New Tobacco Internet Survey that was conducted in February 2021(JASTIS 2021). It was in the third wave of the pandemic and approximately one year after the first patient in Japan. The numbers of subjects were different between the two surveys: n = 26,000 for the JASTIS 2021 and n = 28,000 for the JACSIS. When the survey was conducted was different between the two surveys. We have accordingly revised the discussion so that we discuss whether gender affected the fear and anxiety about and perceived susceptibility to COVID-19 even nearly one year after the onset of the pandemic.

Revision

Materials and Methods, lines 62 – 85

Reviewer 2

Abstract.

  • Authors stated: “It may be universal that women are more anxious and fearful about and perceived susceptibility to infectious diseases”.

I think it is a difficult statement to support. and I also think it is widely disputable to be able to define something potentially universal with such a shallow study. Remaining in doubt as to what might be universal.

Response

As you pointed out, saying “It may be universal …” may sound assertive. We have made it humble by saying “We observed that …”

Revision

Abstract, lines 25 – 27

Discussion, lines 258 – 260

Introduction.

  • The introduction gives some information on existing studies with respect to gender differences in relation to fear of contagion - but does not discuss any other characteristics.  Women are mentioned as a uniform category, failing to get any information - due in an introduction - with respect to other variables (socio-economic status, ethnicity..etc. – intersectionality)
  • The authors then include it in the confounding factors (where they leave out nationality, ethnicity, or religious affiliation..) but do not mention it in the introduction, - nor elsewhere in the text, which would be needed.

Response

As you pointed out, gender difference does not merely mean biological difference. Socioeconomic and health statuses and lifestyles differ by gender and must be considered when examining the effects of gender differences. We have clarified this point in the introduction. You mentioned income later, which is one of such socioeconomic statuses.

We did not collect data on nationality, ethnicity, and religious affiliation. We have mentioned this as a study limitation.

Revision

Introduction, lines 52 – 56

Discussion, lines 247 – 255

  • They say in the introduction that the results can help develop interventions and health policies. On this introduction? Health policy only relative to the issue of being in the woman category? Not including an intersectional lens is a bit anachronistic nowadays. There are many studies, for example done in countries where there is a great deal of discrimination related to class, ethnicity .. (see US or Israel), that have highlighted how these variables meet together in increasing perceived risk/actual risk of infection.

Response

We agree with your suggestion that we cannot speak about health policymaking only based on the present findings. Therefore, we have deleted such a description from the introduction and discussion.

Revision

Introduction (deleted)

This finding contributes to the discussion on whether treatment interventions and health policies must consider gender differences in infectious disease pandemics [3,7,8,13].

Introduction (deleted)

Previous researchers who found the differences between men and women in fear and anxiety about and perceived susceptibility to COVID-19 insisted on considering the gender differences for the treatment interventions, including both physical and mental healthcare and health policies for the infectious disease pandemic [3,8,9,13,14]. Our finding raises this necessity also in Japan, which has a gender imbalance and a small number of women in the political and administrative bureaucracy [29]. Women’s perspectives must be included in health policymaking.

Method and participants.

  • Difficult to express with the fact that the authors say the details are described elsewhere.

Response

We have explained the sampling method in more detail so that we do not need to say “The details were described elsewhere.”

Revision

Materials and Methods (2.2. Subjects), lines 63 – 86

  • Dividing the subjects into two bands, 15-64 and 65-49 is a bit reductive. Wasn't it possible to have a few more bands?

Response

Age has been categorized basically in the 10-year age range.

Revision

Materials and Methods (2.4. Potential Confounding Factors), lines 113 – 114

New Tables 1 – 7 have been modified and made.

  • Does having job information with the required categories (regular, not regular) give us any indication with respect to (in)sufficient income? I guess this could have a major implication in any person concerns about Covid or any other disease. In the US, for instance - being able to have an high salary or not have a major influence on my access to health facilities.. and not just in the US..

Response

We have added household income to the independent variables. Women were significantly associated both with fear and anxiety about COVID-19 and with perceived susceptibility to COVID-19 even after adjustment for household income.

Revision

Introduction, lines 52 – 56

Materials and Methods (2.4. Potential Confounding Factors), lines 110 – 113, 126 – 129

New Tables 1 – 7 have been modified and made.

  • The discussion, like the rest of the work in my opinion is very superficial. “A possible explanation is that women tend to be more concerned about their own and their family’s health and seek medical care more frequently than men” Again, it seems to me that there is an enormity of literature-to date and even in the last 20 years-that exposes how this is not an explanation... What variables come into play for women to be more concerned-or for men to be less concerned-about health? it seems to me that this study doesn't tell us much...

Response

Reviewer 1 also found our discussion superficial. We have improved it as follows:

  • 2nd paragraph: We have shown the advantage of our study that we considered socioeconomic and health statuses and lifestyles that differed by gender in the analyses.
  • 4th paragraph: We have discussed whether the FoCS scores of Japanese general people changed as time went by.
  • 6th paragraph: We have discussed the effect size of age and the potential confounding factors on the FoCS scores and perceived susceptibility to COVID-19.
  • 7th paragraph: We have discussed the study limitations in more detail. One topic is the potential selection biases related to the internet-based survey. Another topic concerns the absence of data on nationality, ethnicity, and religious affiliation. This made us difficult to discuss the possibility of cultural differences.

We have deleted the sentence “A possible explanation is that women tend to be more concerned about their own and their family’s health and seek medical care more frequently than men.” In the revised manuscript, we have made our discussion based on the present findings rather than simply citing the previous literature.

We would be glad if you kindly evaluate our revised discussion.

Revision

Discussion, lines 247 – 256, 268 – 279, 287 – 295, and 296 - 313

Discussion (deleted)

A possible explanation is that women tend to be more concerned about their own and their family’s health and seek medical care more frequently than men.

Round 2

Reviewer 1 Report

I appreciate the authors' efforts to address my review comments. The addition of potential confounding factors such as noncommunicable diseases, age, employment status, etc. really helps to put the effect size of gender into context. It's very interesting that the effect of gender, though small, is larger than the effect of factors that objectively increase the risk of COVID-19 (such as having other illnesses, being older). 

I think the authors could do more to draw out some of the interesting patterns in the results for the reader (in the discussion section) - for instance, older people report reduced susceptibility to COVID-19 compared to young people and only slightly increased fear - could it be that older people were successfully managing their fear of COVID-19 by taking precautions (e.g. social distancing) and that this made them feel that they were not going to get COVID-19 and hence did not need to fear it? This is speculation, of course, but I think there is plenty of material for the discussion that is still sort of left to the reader to discover on their own.

On the whole, though, I am satisfied that the authors have adequately addressed my comments.

Author Response

Reply to the review comments

We appreciate the editor for the opportunity to submit the revised manuscript. We also appreciate Reviewer 1 for the high evaluation and considerate comments. Below is our reply. We highlighted our revision in yellow in our revised manuscript.

Correction

In addition to our responses to Reviewer 1’s comments, we have made the following correction.

  1. Abstract, lines 21 – 22: We have added the effect size for the association between gender and perceived susceptibility to COVID-19.
    • perceived susceptibility (13.6% vs. 11.5%, φ = 0.032)
  2. Reference no. 22 was written in Japanese. We have added the phrase “in Japanese” to it.

Reviewer 1

  • I appreciate the authors' efforts to address my review comments. The addition of potential confounding factors such as noncommunicable diseases, age, employment status, etc. really helps to put the effect size of gender into context. It's very interesting that the effect of gender, though small, is larger than the effect of factors that objectively increase the risk of COVID-19 (such as having other illnesses, being older).
  • On the whole, though, I am satisfied that the authors have adequately addressed my comments.

Response

We appreciate your high evaluation of our paper.

  • I think the authors could do more to draw out some of the interesting patterns in the results for the reader (in the discussion section) - for instance, older people report reduced susceptibility to COVID-19 compared to young people and only slightly increased fear - could it be that older people were successfully managing their fear of COVID-19 by taking precautions (e.g. social distancing) and that this made them feel that they were not going to get COVID-19 and hence did not need to fear it? This is speculation, of course, but I think there is plenty of material for the discussion that is still sort of left to the reader to discover on their own.

Response

We have added a paragraph to discuss age, having a spouse, and employment status (not working) whose associations with the Fear of COVID-19 Scale (FoCS) scores were different from those with perceived susceptibility to COVID-19.

We are afraid we did not include social distancing in the discussion. We believe that social distancing would be a consequence of fear and anxiety about and perceived susceptibility to COVID-19, while gender, age, and other independent variables we employed were potential causes of them. Therefore, the inclusion of social distancing in the analysis avoids the logical interpretation of the results.

However, for your reference, we conducted analyses that included social distancing. The results are shown in Tables A – G (see the PDF file attached to this reply). Women remained significant regarding the associations with FoCS scores and perceived susceptibility to COVID-19. We asked the subjects in our survey how often they were paying attention to social distancing, i.e., the practice of maintaining a physical distance of 2 meters or longer from other people.

Revision

Discussion, lines 294 – 307

We here discuss age, having a spouse, and employment status (not working) whose associations with FoCS scores were different from those with perceived susceptibility to COVID-19. It was not clearly shown that FoCS scores increased with age. Having a spouse was significantly associated with high FoCS scores even after adjustment for age and other potential confounding factors. On the other hand, perceived susceptibility to COVID-19 was inversely associated with age. A similar finding was observed in a study in Israel [17]. The association between having a spouse and perceived susceptibility to COVID-19 was not significant in the multivariable analysis. This would suggest that having a spouse was a confounding factor for the association between age and perceived susceptibility to COVID-19. In our study, the prevalence of having a spouse was higher in the older age than in the younger age (Supplementary Table 8). Not working was not associated with FoCS scores but with perceived susceptibility to COVID-19. Compared to those who were working, those who were not working would have fewer opportunities to meet other people outside, leading to less perceived susceptibility to COVID-19.

Supplementary Table 8 “Age and having a spouse (n = 23,516)” has been newly made.
